# Enhanced Photodegradation of p-Nitrobenzoic Acid by Binary Mixtures with Ba^2+^/TiO_2_ and MCM-41

**DOI:** 10.3390/ma14092404

**Published:** 2021-05-05

**Authors:** Xianyuan Fan, Hong Liu, Weikun Song, Chia-Yuan Chang

**Affiliations:** 1Department of Resource and Environmental Engineering, Wuhan University of Science and Technology, Wuhan 430081, China; fanxianyuan@wust.edu.cn; 2Hubei Key Laboratory for Efficient Utilization and Agglomeration of Metallurgic Mineral Resources, Wuhan University of Science and Technology, Wuhan 430081, China; 3China Institute of Water Resources and Hydropower Research, Beijing 100048, China; songwk@iwhr.com; 4Department of Environment Engineering and Science, Chia Nan University of Pharmacy and Science, Tainan 71710, Taiwan; fanxy1027@126.com

**Keywords:** TiO_2_, Ba(II), MCM-41, p-nitrobenzoic acid, photodegradation

## Abstract

A novel Ba(II)/TiO_2_–MCM-41 composite was synthesized using binary mixtures with Ba^2+^/TiO_2_ and MCM-41, and Ba^2+^ as a doping ion of TiO_2_. The specific surface area and pore structure characterizations confirm that a mesoporous structure with a surface area of 341.2 m^2^/g and a narrow pore size distribution ranging from 2 to 4 nm was achieved using Ba(II)/TiO_2_–MCM-41. Ba(II)/TiO_2_ particles were synthesized into 10–15 nm particles and were well dispersed onto MCM-41. The diffraction peaks in the XRD patterns of TiO_2_–MCM-41 and Ba(II)/TiO_2_–MCM-41 were all attributed to anatase TiO_2_. By taking advantage of MCM-41 and Ba^2+^, the photocatalytic performance of Ba(II)/TiO_2_–MCM-41 was remarkably enhanced by suppressing its rutile phase, by lowering the band gap energy, and by facilitating the dispersion of TiO_2_. Therefore, the photodegradation efficiencies of p-nitrobenzoic acid (4 × 10^−4^ mol/L) by various photocatalysts (60 min) under UV light irradiation are arranged in the following order: Ba(II)/TiO_2_–MCM-41 (91.7%) > P25 (86.3%) > TiO_2_–MCM-41 (80.6%) > Ba(II)/TiO_2_ (55.7%) > TiO_2_ (53.9%). The Ba(II)/TiO_2_–MCM-41 composite was reused for five cycles and maintained a high catalytic activity (73%).

## 1. Introduction

TiO_2_ has been extensively studied as a photocatalyst in the photodegradation of organic pollutants due to its relatively high photocatalytic activity, excellent chemical stability, and nontoxic characteristics [1,2,3,4]. However, its associated disadvantages, such as low quantum yield and relatively small surface area, as well as its difficulty being separated and retrieved from treated water, may limit its in situ application in photodegrading organic pollutants [5,6]. In addition, TiO_2_ nanoparticles are small but easily form an agglomerate. During the past several decades, researchers have discovered that the agglomeration of TiO_2_ nanoparticles and their phase transition from an anatase to a rutile form can be inhibited efficiently [4,5]. Thus, to address these issues, improvements have been conducted by using surface modification, porous material loading or blocking, and metal/nonmetal doping [7,8,9,10].

Metal doping provides a convenient method of solving the problems associated with the use of TiO_2_ in photodegradation [11,12,13]. Both the photophysical and the photochemical activities of TiO_2_ are affected by the doping of metal ion and oxide [14,15]. Transition metals (such as Cr [16], Ag [17], Fe [18], Rh [19], and V [20]) have been employed as dopants of TiO_2_, but the phenomenon with alkaline earth metals is understudied [21]. Kumaresan [22] prepared TiO_2_ nanoplates doped with Sr^2+^ and evaluated their photocatalytic activities using 2,4-dinitrophenol (2,4-DNP) as a target pollutant. The complete mineralization of 2,4-DNP by Sr^2+^-doped TiO_2_ nanoplates required a shorter photodegradation time than the TiO_2_ nanoparticles. According to Rodriguez et al. [23], the ionic radius of dopants and their electric charges are the important parameters that control stabilization of the anatase phase, and the transition temperature of the anatase–rutile process is plotted as a function of the charge of the ion times its volume. Ba^2+^ demonstrated the best stabilization due to its largest charge ionic volume.

Since Mobil Company successfully synthesized an MCM-41 molecular sieve (MCM-41) in 1992, MCM-41 has been widely used in catalysis, adsorption, separation, and other fields because of its large specific surface area, high porosity, narrow pore size distribution, and other beneficial characteristics [24,25,26]. Although UV and visible light are not absorbed by MCM-41 composed of SiO_2_, its readily accessible mesoporous frameworks and adjustable porosities are beneficial to the incorporation of metal ions [27,28]. In addition, by dispersing photocatalysts on MCM-41, the composites are provided with improved light absorption performance [28,29]_._ According to Dong et al. [29], the photocatalytic efficiency of Acid Red B by TiO_2_/MCM-41 reached 100% within 2 h.

In this study, the feasibility and role of Ba^2+^ as a doping ion of TiO_2_ were evaluated. The mesoporous MCM-41 was mechanically mixed with the Ba(II)/TiO_2_ composite to disperse and improve the reactive surface area of the aqueous media. The physical and chemical properties of Ba(II)/TiO_2_–MCM-41 were characterized by adopting a specific surface analyzer, a transmission electron microscope (TEM), X-ray photoelectron spectroscopy (XPS), X-ray diffraction (XRD), and UV-vis diffuse reflectance spectroscopy (UV-vis DRS). The photocatalytic activity and reaction rate constant of Ba(II)/TiO_2_–MCM-41 was evaluated by a photodegrading p-nitrobenzoic acid in an aqueous solution under UV light irradiation. The findings obtained were compared with that of the TiO_2_–MCM-41, Ba(II)/TiO_2_, TiO_2_, and P25. Finally, the service life and cycle times of the Ba(II)/TiO_2_–MCM-41 catalyst were investigated.

## 2. Materials and Methods

### 2.1. Materials and Chemicals

MCM-41, Degussa P25 (P25), was obtained from Nankai University Catalyst Co. (Tianjin, China). Tetrabutyl titanate (Ti(OBu)_4_), Ba(NO_3_)_2_, anhydrous alcohol, acetic acid (99.5%), and p-nitrobenzoic acid were purchased from Tianda Chemical Co. (Tianjin, China). All chemicals used in this study were of analytical grade and were used as received.

### 2.2. Preparation of Ba(II)/TiO_2_–MCM-41

#### 2.2.1. Preparation of Ba(II)/TiO_2_ and TiO_2_

The conventional sol–gel method was employed to prepare the Ba(II)/TiO_2_ composite. Specifically, 0.0028 g Ba(NO_3_)_2_ was dissolved in 8.75 mL ethanol and, then 0.5 mL acetic acid (99.5%) was added. The resulting solution was stirred for 10 min at 250 rpm. Subsequently, the mixture of 8.75 mL ethanol and 7.5 mL Ti(OBu)_4_ was added dropwise into the solution. The mixture was continuously stirred until the gel was formed. The gel was dried at 80 °C and ground after cooling. Finally, a calcination kept at 700 °C for 2 h was carried out to obtain the Ba(II)/TiO_2_ composite. The mass ratio of TiO_2_:Ba (II) in the resulting composite was 3:0.0025. Same procedure was adopted for TiO_2_ preparation except for the addition of Ba(NO_3_)_2_.

#### 2.2.2. Preparation of Mixtures with Ba(II)/TiO_2_ and MCM-41

The binary mixtures with Ba(II)/TiO_2_ and MCM-41 were prepared by a liquid phase mixing method. First, 0.0028 g Ba(NO_3_)_2_ was dissolved in 8.75 mL ethanol, after which 0.5 mL acetic acid (99.5%) was added to the solution and stirred for 10 min at 250 rpm. Subsequently, the mixture of 8.75 mL ethanol and 7.5 mL Ti(OBu)_4_ was added dropwise into the solution. After the solution was stirred rapidly for 30 min, 1.2 g of MCM-41 powder was mixed. Next, 3 mL of deionized water was added dropwise under vigorous stirring. The mixture was continuously stirred until the gel was formed. The gel was dried at 80 °C and then ground after cooling down. Finally, a calcination kept at 700 °C for 2 h was carried out to obtain the Ba(II)/TiO_2_–MCM-41 composite. The mass ratio of the Ba(II)/TiO_2_ to MCM-41 in the resulting composite was 3:2. The same procedure was adopted for TiO_2_–MCM-41 preparation, except for the addition of Ba(NO_3_)_2_.

### 2.3. Characterizations

The X-ray diffraction (XRD) measurements for the Ba-doped and the undoped TiO_2_–MCM-41 powders were conducted by an X-ray diffractometer (max-IIIA, Rigaku D, Japan). The specific surface area of the N_2_ adsorption–desorption isotherm and the pore parameters were measured using an Autosorb-1-MP/LP analyzer (Quantachrome instruments, Tallahassee, FL, USA). A morphological analysis was carried out by an H-9500 transmission electron microscope (Hitachi, Hitachi, Japan). The UV-vis absorption spectrum and the absorbance (λ = 270 nm) of p-nitrobenzoic acid at different photodegradation times were obtained using a UV-2550 spectrophotometer (Shimadzu, Tokyo, Japan). The UV-vis diffuse reflectance spectra of the Ba(II)/TiO_2_–MCM-41 were also obtained by using UV-2550 with the aid of a solid module. The surface chemical compositions and oxidation states were examined using an X-ray photoelectron spectrometer (Escalab 250Xi, Thermo Fisher Scientific, Waltham, MA, USA).

### 2.4. Photocatalytic Activity Measurement

Photodegradation of p-nitrobenzoic acid by different photocatalysts was conducted in a reactor equipped with a 300 W high-pressure mercury lamp emitting UV irradiation mainly in the range of 250–400 nm [30]. The photocatalyst was dispersed into the p-nitrobenzoic acid (4 × 10^−4^ mol/L, pH = 4.0 ± 0.1) aqueous solution at a dose of 0.5 g/L, and the mixture was stirred at 200 rpm for 60 min. At given time intervals, a 10 mL suspension was withdrawn and centrifuged, and the supernatant was used to determine the concentration of p-nitrobenzoic acid after UV irradiation.

The photodegradation efficiency of p-nitrobenzoic acid can be expressed as follows: η = (C_0_ − C)/C_0_ × 100% = (A_0_ − A)/A_0_ × 100%(1)
where C_0_ and A_0_ are the initial concentration and the absorbency of p-nitrobenzoic acid, respectively, and where C and A are the concentration and absorbency of p-nitrobenzoic acid after photodegradation time (t), respectively.

## 3. Results

### 3.1. BET Specific Surface Area and Pore Parameters

The pore size distribution and the N_2_ adsorption–desorption isotherms of the Ba(II)/TiO_2_–MCM-41 are shown in Figure 1. It was found that the Ba(II)/TiO_2_–MCM-41 composite exhibits a narrow pore size distribution of 2–4 nm with a characteristic peak at 2.71 nm.

The BET (Brunauer-Emmett-Teller) specific surface area (S_BET_) and the pore parameters of TiO_2_, MCM-41, TiO_2_–MCM-41, and Ba(II)/TiO_2_–MCM-41 are presented in Table 1. The S_BET_ of the Ba(II)/TiO_2_–MCM-41 composite was 341.2 m^2^·g^−1^, and this was found to be lower than that of MCM-41 (805.8 m^2^·g^−1^) due to loading of Ba(II)/TiO_2_ onto the surface of MCM-41 but was much higher than that of TiO_2_ (65.8 m^2^·g^−1^). The large S_BET_ of Ba(II)/TiO_2_–MCM-41 could enhance the contact with organic molecules and increase the adsorption of water and hydroxyl. The adsorbed water and hydroxyl may react with the photoexcited holes on the surface of Ba(II)/TiO_2_–MCM-41 to generate hydroxyl radicals. In addition, the mesoporous structure and high pore volume (0.357 cm^3^/g) of the Ba(II)/TiO_2_–MCM-41 composite allowed for rapid diffusion of the reactants and products during the photocatalytic reaction. These properties of Ba(II)/TiO_2_–MCM-41 are beneficial to the photodegradation of organic contaminants.

### 3.2. TEM Images

The TEM images of MCM-41 and Ba(II)/TiO_2_–MCM-41 are shown in Figure 2. As shown in Figure 2a, the MCM-41 is characterized by a highly ordered structure with a uniform aperture. Figure 2b shows that the Ba(II)/TiO_2_ particles are well dispersed on the surface of the MCM-41, and this may be attributed to the fact that MCM-41 is a mesoporous material with a pore size of 3.15 nm (as shown in Table 1). In addition, the Ba(II)/TiO_2_ particles are too large (5–15 nm) to enter the pores of the MCM-41. As a result, the Ba(II)/TiO_2_–MCM-41 composite exhibits less regularity in pore structure. However, the mesoporous structure is still maintained.

### 3.3. XPS Analyses

Figure 3 presents a typical survey spectrum of Ti 2p, O 1s, and Si 2p. It can be observed that Ti and O show the strongest peaks, as expected since they make up the crystal lattice of TiO_2_. Unfortunately, the characteristic peak of Ba was not observed, which may be due to the low content of Ba in the Ba(II)/TiO_2_–MCM-41 composite. In order to determine the chemical state of Ti/Si in the composite, the high-resolution XPS spectra of Ti, Si, and O were analyzed separately, as shown in Figure 4. The XPS spectra of the Ti 2p spin orbit components were located at bonding energies of 458.8 and 464.3 eV, thus corresponding to Ti 2p3/2 and Ti 2p1/2, respectively. The strongest peak is the Ti 2p3/2, and this corresponds with tetravalent Ti^4+^. These results are in accordance with that in the literature [31,32].

The XPS spectrum of the Si 2p includes two peaks located at 102.8 and 102.2 eV that are attributed to the Si–O–Si and Ti–O–Si linkages [33]. This evidence confirms that Ti^4+^ in a titania octahedral lattice can be replaced by Si^4+^, and vice versa [34,35]. The XPS spectrum of the O 1s comprises several peaks in the regions from 528 to 537 eV. The strong peaks of O 1s appear at 529.8 and 533.2 eV and are ascribed to oxygen in the Ti–O–Ti linkage of the TiO_2_ and OH groups in Ti–OH and Si–OH bonding, respectively. The peak arising at 530.7 eV belongs to Ti–O–Si bonds, and this could be due to the coordination of Ti and Si atoms in the matrix surface [32].

### 3.4. XRD Analyses

The XRD patterns of MCM-41, TiO_2_, TiO_2_–MCM-41, and Ba(II)/TiO_2_–MCM-41 are shown in Figure 5. The peaks at 2θ = 5.7° (Figure 5b) and 23.0° (Figure 5a) are characteristic of MCM-41, as reported elsewhere [28,29,36,37]. The diffraction peaks of the anatase phase and rutile phase appear simultaneously in the diffraction pattern of TiO_2_. There are ten obvious rutile characteristic peaks (2θ = 27.5°, 36.1°, 39.3°, 41.2°, 44.1°, 54.2°, 56.8°, 64.2°, 69.0°, and 69.9°), while only two anatase characteristic peaks can be observed. However, the XRD pattern of TiO_2_–MCM-41 shows an opposite situation, with all of the diffraction peaks being attributed to the anatase phases. It was reported that TiO_2_ will be transformed from an anatase phase to a rutile phase when the calcination temperature is higher than 600 °C [29]. The TiO_2_ in TiO_2_–MCM-41 still exists in an anatase phase under 700 °C calcination for 2 h in the present study, and this may be caused by the formation of a Ti–O–Si bond between the Ti (Ⅳ) and Si (Ⅳ). In other words, the loading of TiO_2_ onto the surface of MCM-41 inhibits the phase transformation of TiO_2_ at higher calcination temperatures.

The doping of Ba^2+^ into TiO_2_–MCM-41 has no obvious effect on crystal structure, except that the diffraction peak intensity of the anatase phase is slightly higher. This is because the radius of Ba^2+^ (134 pm) is larger than that of Ti^4+^ (61 pm) and close to the radius of O^2−^ (140 pm). The anatase structure is more open than the rutile structure, and large substituting groups can be more easily accommodated in this structure than in a rutile structure [23]. As a result, the anatase crystalline structure is more difficult to transform into a rutile structure.

The calculated diffraction patterns of the Ba(II)/TiO_2_–MCM-41 composite were analyzed using the Scherrer equation to infer the crystallite size, ≈33.5 nm. The particle size measured via TEM is smaller (10–15 nm) than that deduced from the Scherrer formula (≈33.5 nm, Figure 2b) because the fusion of the neighboring tiny particles caused substantial overestimation of the primary particles, as suggested recently by Weidenthaler [38].

### 3.5. UV-Vis Diffuse Reflectance Spectra

The UV-visible diffuse reflectance spectra of TiO_2_, TiO_2_–MCM-41, Ba(II)/TiO_2_–MCM-41, and P25 are presented in Figure 6. The Ba(II)/TiO_2_–MCM-41 composite exhibited a higher absorbance than the other photocatalysts in the UV light range. Moreover, the absorption spectra of Ba(II)/TiO_2_–MCM-41 is slightly shifted to the visible light range. According to the equation Eg = 1240/λ (where λ is the wavelength edge of the absorption band) [39], the band gaps energy of TiO_2_, TiO_2_–MCM-41, P25, and Ba(II)/TiO_2_–MCM-41 were calculated as 3.26, 3.22, 3.21, and 3.19 eV, respectively. The decrease in band gap energy increased the number of photogenerated electrons and holes to participate in the photocatalytic reaction. This resulted in an improvement in the photocatalytic activity of Ba(II)/TiO_2_–MCM-41.

### 3.6. Photodegradation of p-Nitrobenzoic Acid

The photocatalytic reactivity of the Ba(II)/TiO_2_–MCM-41 composite was evaluated by the photodegradation of p-nitrobenzoic acid under UV light irradiation and compared with that of TiO_2_–MCM-41, Ba(II)/TiO_2_, TiO_2_, and P25 (shown in Figure 7a). Moreover, the photodegradation efficiency of p-nitrobenzoic acid under only UV irradiation (without photocatalyst) and the adsorption efficiency of MCM-41 and Ba(II)/TiO_2_–MCM-41 on p-nitrobenzoic acid without irradiation were also tested (shown in Figure 7b). The photodegradation efficiencies of all photocatalysts are as follows: Ba(II)/TiO_2_–MCM-41 > P25 > TiO_2_–MCM-41 > Ba(II)/TiO_2_ > TiO_2_. The highest photocatalytic activity belongs to Ba(II)/TiO_2_–MCM-41, with a photodegradation efficiency of 91.7% at 60 min. The following photodegradation efficiencies, 86.3%, 80.6%, 55.7%, and 53.9%, were achieved by P25, TiO_2_–MCM-41, Ba(II)/TiO_2,_ and TiO_2_, respectively. These results indicate that the photocatalytic activity of TiO_2_ is enhanced by the addition of MCM-41 and by doping Ba(II), hence the higher photocatalytic activity than that of P25.

Due to the contributions of both photocatalysis and the adsorption towards the removal of p-nitrobenzoic acid, the adsorption ability of Ba(II)/TiO_2_-MCM-41 and MCM-41 were characterized in a dark condition (no irradiation) as shown in Figure 7b. The adsorption efficiency of Ba(II)/TiO_2_–MCM-41 after 60 min was mainly due to the MCM-41.

MCM-41 prevented the anatase from transforming into a low activity rutile phase, and the mesoporous structure and facilitated dispersion of MCM-41 allowed for rapid contact and diffusion of the reactants and products. The lower band gap energy of Ba(II)/TiO_2_–MCM-41 accelerated the generation of photogenerated electrons and holes in the photocatalytic reaction, and consequently assisted with the improvement in the photocatalytic activity of Ba(II)/TiO_2_–MCM-41.

The photodegradation kinetics of p-nitrobenzoic acid can be described by the zero order equation as follows:(2)$$C_{0}-C=\mathrm{kt}$$
where C_0_ and C are the initial concentration and the residue concentration after irradiation of p-nitrobenzoic acid solution, respectively; t is the time for irradiation (min); and k is the kinetic rate constant obtained from the slope of the linear graph illustrated in Figure 8. All graphs are linear, with a coefficient of determination (R^2^) greater than 0.99. The reaction rate constants (k) and R^2^ are summarized in Table 2.

Among various photocatalysts, the reaction rate constant of the Ba(II)/TiO_2_–MCM-41 is the highest value of 0.00631 mmol·L^−1^·min^−1^. Therefore, the mesoporous structure and facilitated dispersion of the Ba(II)/TiO_2_/MCM-41 allowed for rapid contact and diffusion of the reactants and products, which consequently enhanced the efficiency and reaction rate of the photocatalytic reaction.

In addition to the photocatalytic activity, stability was another important issue for their practical application. The recycling test for the photocatalysis of p-nitrobenzoic acid are shown in Figure 9. In our experiment, the stability of the Ba(II)/TiO_2_–MCM-41 composite was evaluated by performing the cycling experiments under the same conditions. With the reuse time increasing, the photocatalytic activity of p-nitrobenzoic acid demonstrated no obvious reduction. The Ba(II)/TiO_2_–MCM-41 composite reused for five cycles maintained a high catalytic activity (about 73%). This result indicated that the Ba(II)/TiO_2_–MCM-41 composite exhibited an excellent recycle performance.

## 4. Conclusions

This study investigated the physical and chemical properties of TiO_2_ doping with Ba^2+^ and mixing with MCM-41 as well as the photocatalytic activity of the Ba(II)/TiO_2_–MCM-41 composite in the photodegrading p-nitrobenzoic acid under UV light irradiation. It was found that the BET specific surface area of Ba(II)/TiO_2_–MCM-41 (341.2 m^2^·g^−1^) is much larger than that of TiO_2_ (65.8 m^2^·g^−1^). The TEM spectra showed that the TiO_2_ and Ba(II)/TiO_2_ particles were synthesized into 10–15 nm particles and well dispersed onto the MCM-41. The XPS spectra showed the formation of Ti–O–Si bonds in Ba(II)/TiO_2_–MCM-41, which inhibited the transformation of TiO_2_ from an anatase phase to a rutile phase. As a result, all of the diffraction peaks in the XRD patterns of TiO_2_–MCM-41 and Ba(II)/TiO_2_–MCM-41 were attributed to the anatase TiO_2_. Among the five photocatalysts (TiO_2_, Ba(II)/TiO_2_, TiO_2_–MCM-41, Ba(II)/TiO_2_–MCM-41, and P25), the photocatalytic performance of Ba(II)/TiO_2_–MCM-41 was remarkably enhanced by suppressing the rutile phase, by lowering the band gap energy, and by facilitating the dispersion of TiO_2_ by using the advantages of MCM-41 and Ba^2+^. The efficiency of the photodegrading p-nitrobenzoic acid (4 × 10^−4^ mol/L) under UV light irradiation by Ba(II)/TiO_2_–MCM-41 reached 91.7% at 60 min. The Ba(II)/TiO_2_–MCM-41 composite was reused for five cycles and maintained a high catalytic activity, thus indicating that the Ba(II)/TiO_2_–MCM-41 composite exhibited an excellent recycle performance.

## Figures and Tables

**Figure 1 materials-14-02404-f001:**
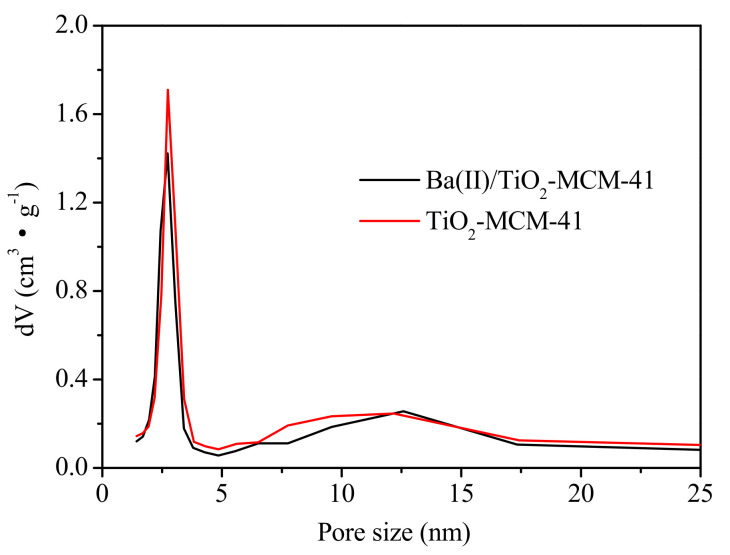
Pore size distribution of Ba(II)/TiO_2_–MCM-41 and TiO_2_–MCM-41.

**Figure 2 materials-14-02404-f002:**
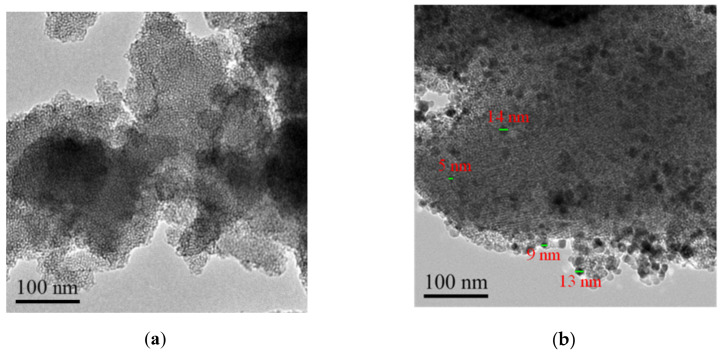
TEM images of MCM-41 (**a**) and Ba(II)/TiO_2_/MCM-41 (**b**).

**Figure 3 materials-14-02404-f003:**
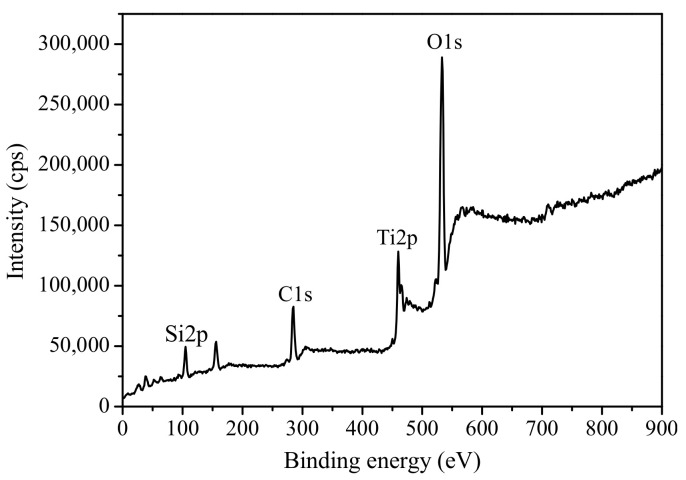
XPS survey spectra of the Ba(II)/TiO_2_–MCM-41 composite.

**Figure 4 materials-14-02404-f004:**
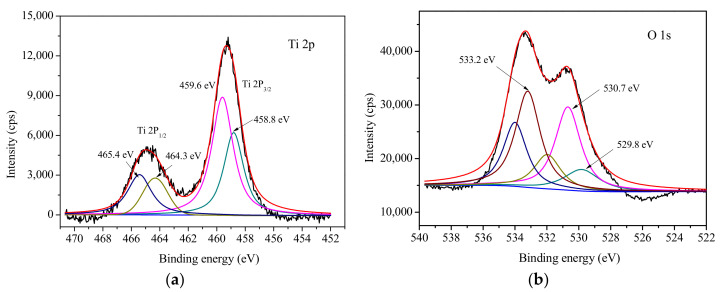
High-resolution XPS spectra of the Ba/TiO_2_–MCM-41composite for Ti 2p (**a**), O 1s (**b**), and Si 2p (**c**).

**Figure 5 materials-14-02404-f005:**
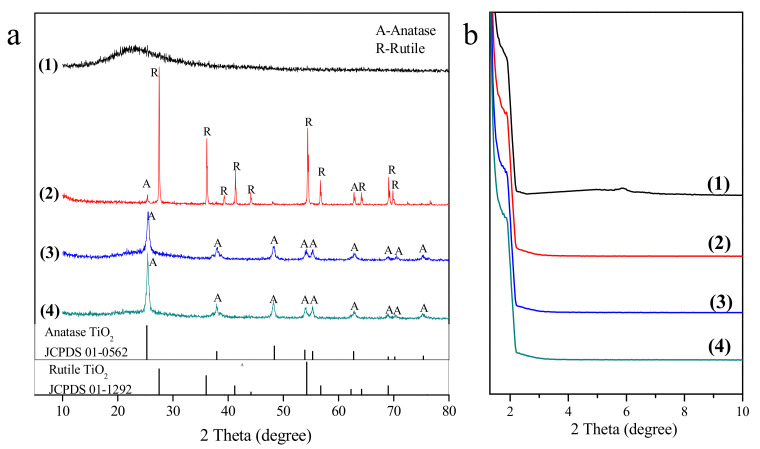
XRD patterns of MCM-41—1, TiO_2_—2, TiO_2_–MCM-41—3, and Ba(II)/TiO_2_–MCM-41—4 scanned at wide angle (**a**) and low angle (**b**).

**Figure 6 materials-14-02404-f006:**
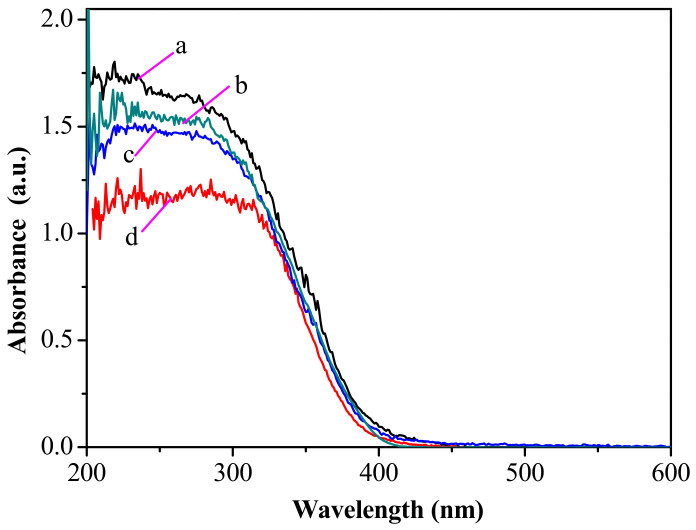
UV-vis diffuse reflectance spectra of Ba(II)/TiO_2_–MCM-41—a, P25—b, TiO_2_–MCM-41—c, and TiO_2_—d.

**Figure 7 materials-14-02404-f007:**
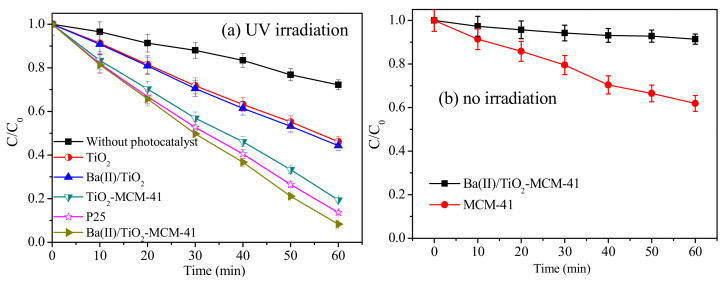
Photodegradation efficiencies of p-nitrobenzoic acid by different photocatalysts under (**a**) UV irradiation and (**b**) no irradiation (reaction conditions: C_0_ = 4 × 10^−4^ mol/L, dosage of material = 0.5 g/L, initial pH = 4.0 ± 0.1).

**Figure 8 materials-14-02404-f008:**
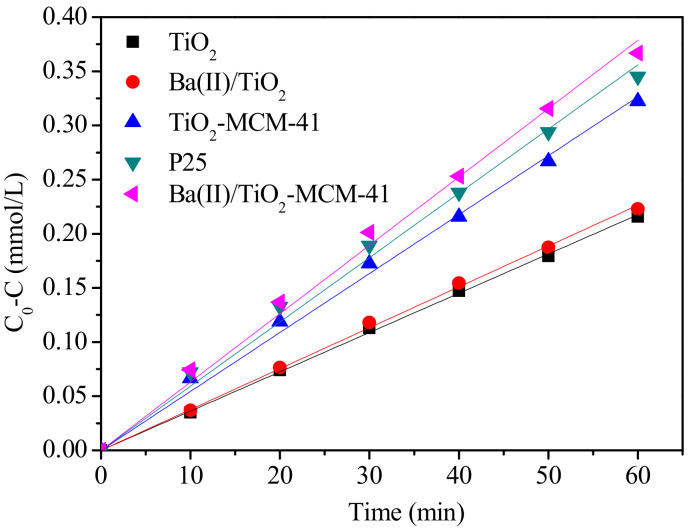
Zero order models for the photodegradation of p-nitrobenzoic acid by various photocatalysts.

**Figure 9 materials-14-02404-f009:**
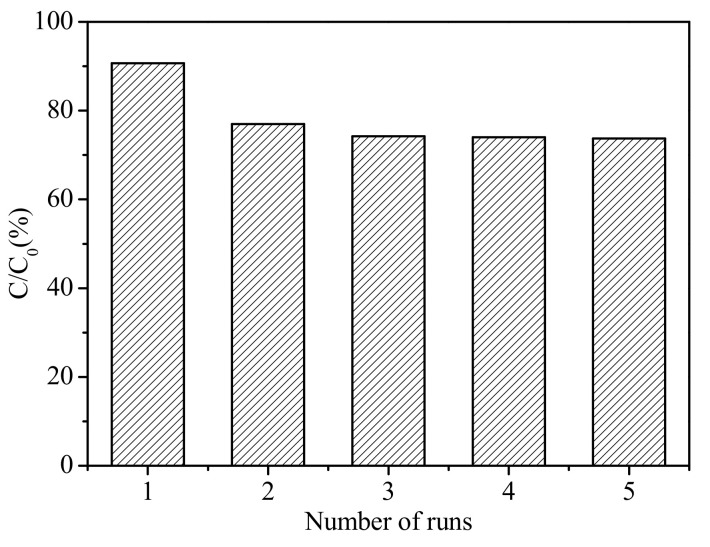
Recycling test for the photocatalysis of p-nitrobenzoic acid by Ba(II)/TiO_2_–MCM-41 (reaction conditions: C_0_ = 4×10^-4^ mol/L, dosage of material = 0.5 g/L, reaction time: t = 60 min, initial pH = 4.0 ± 0.1).

**Table 1 materials-14-02404-t001:** S_BET_ and pore parameters of different materials.

Samples	Specific Surface Area (m^2^·g^−1^)	Average Pore Size (nm)	Pore Volume (mL·g^−1^)
TiO_2_	65.8	3.75	0.207
MCM-41	805.8	3.15	0.811
TiO_2_–MCM-41	358.8	2.75	0.429
Ba(II)/TiO_2_–MCM-41	341.2	2.71	0.357

**Table 2 materials-14-02404-t002:** Reaction rate constant (k) and correlation coefficient (R^2^) of various photocatalysts.

Photocatalysts	k (mmol·L^−1^·min^−1^)	R^2^
TiO_2_	0.00360	0.999
Ba(II)/TiO_2_	0.00377	0.999
P25	0.00593	0.997
TiO_2_–MCM-41	0.00544	0.998
Ba(II)/TiO_2_–MCM-41	0.00631	0.998

## Data Availability

All relevant data presented in the article are stored according to institutional requirements and, as such, are not available online. However, all data used in this manuscript can be made available upon request to the authors.

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
