# Peer review of "Enhanced Photodegradation of p-Nitrobenzoic Acid by Binary Mixtures with Ba2+/TiO2 and MCM-41"

_materials, 2021, doi:10.3390/ma14092404_

Round 1

Reviewer 1 Report

This manuscript proposed by Fan Xianyuan and co-workers describes the possibility to synthesize a Ba(II)/TiO 2 /MCM-41 composite.

X-ray diffraction, UV-vis in diffuse reflectance, TEM and XPS analysis were used in order to characterize the produced composite material. Furthermore, the authors compared the photodegradation of 2,4-dinitrophenol by different materials (TiO2, TiO2 /MCM-4, Ba(II)/TiO2 /MCM-41  and  P25).

The manuscript was not fluidly written, in particular introduction section which results unclear for the reader.

The concentration of acetic acid used during the synthesis procedure is not reported in the manuscript.

The authors do not explain the relevance of the use of MCM-41 in the composite material.

The caption of the figure 4 is not clear.

Did the authors synthetize the TiO2 used in the experimental procedure?

The authors don’t explain thoroughly the reason of the best efficiency of the synthetized Ba(II)/TiO2 /MCM-41 compound.

Photo degradation tests also should be treated for the Ba(II)TiO2 compound.

In the work different unclear sentences and typing errors are present, more in particular in the lines: 37-38; 47; 56; 77; 85; 210.

Experimental data and results need to be discussed deeper; more work needs to be done in order to support the authors’ idea. For all these reasons I will not recommend the manuscript to be accepted for publication.

Reviewer 2 Report

The manuscript deals with the preparation, characterization and testing of novel TiO2-based photocatalysts. Modifications are required before acceptance, according to the comments below.

  • It is difficult to consider MCM-41 as a support in the prepared catalysts, while it seems more a spacer. There are several clues sustaining this hypothesis. First, the mass ratio (TiO2/MCM-41 = 3/2) is not typical of a supported catalyst; TEM images clearly show two phases as XRD; BET surface areas of TiO2/MCM-41 and Ba(II)/TiO2/MCM-41 correspond to about the weight average between those of TiO2 and MCM-41. Accordingly, which is the role of mesoporosity? As a matter of fact, TiO2 (doped or undoped) is not deposed inside the mesopores.
  • About XRD, intensity can depend on the amount of sample loaded and, thus, it is not a significant parameter. FWHM must be evaluated in order to calculate crystallite size by Scherrer equation.
  • In Figure 7 it clearly appears that conversion increases linearly with the reaction time. This is characteristic of zero-order reaction rate rather than first-order. As a matter of fact, despite the good values of R2, the fitting is not good, as figure 8 suggests (all points in the middle above the straight line, all points at the ends below the straight line).
  • There are typos and errors. Please proofread the manuscript carefully.

Reviewer 3 Report

The manuscript (Enhanced photodegradation of p-nitrobenzoic acid by TiO2 loading on MCM-41 and doping with Ba2+) presents interesting results on photodegradation of p-nitrobenzoic acid on different MCM-41 based materials. In my opinion, Authors have done an effort studying the photocatalytic performance of these type of catalysts.

Nevertheless, there are still some doubts or ambiguity which should be clarified before publication.

p.3 line 115-116 – authors write – “the adsorption-desorption isotherms formed a worm-like hysteresis loop, which is a typical characteristic of mesoporous materials”. For MCM-41 typical hysteresis loop is at p/p0~0.4, not at p/p0~0.8. N2 adsorption-desorption isotherms presented by the authors exhibit the characteristic steep increase at around p/p0=0.4, but not hysteresis loop – Fig 1. The presented description is unclear.

Figure 1 inset - Why does the adsorption curve does not cover with the desorption curve below p/p0=0.4. This indicates an incorrected done measurement. The authors should compare N2 adsorption-desorption isotherms of MCM-41, TiO2/MCM-41 and Ba(II)/TiO2/MCM-41 materials.

p.5 lines 167-169 - XRD patterns of TiO2/MCM-41 and Ba(II)/TiO2/MCM-41 at 2θ below 10 degree should be also presented on Figure 2B.

Photocatalytic activity - It is necessary to investigate the service life and cycle times of the Ba(II)/TiO2/MCM-41 catalyst.

p.4 lines 136-140 - The authors write about the mesoporous structure of Ba(II)/TiO2/MCM-41 material and that Ba(II)/TiO2 particles are well dispersed on the surface of MCM-41. On the other hand they claim that the highest rate constant of Ba(II)/TiO2/MCM-41 for photodegradation of p-nitrobenzoic acid is the result of the combined effect of incorporation of Ba(II) and MCM-41 into crystalline structure of TiO2 – p7 lines 231-234. – What is the correct support in this material? So the photodegradation activity part should be clarified with clear explanation by mentioning whether textural property and/or pore structure play a role for the studied reaction. What is the role of Ba(II)? Authors must solve this problem.

Overall, I find this work interesting, thus, I recommend the publication of this manuscript after minor corrections.

Round 2

Reviewer 1 Report

The authors answered satisfactorily all the questions asked. I suggest to add some references in the introduction section: “Chemistry - A European Journal, 2019, 25(62), pp. 14123–14132”, “Dalton Trans., 2020, 49, 17725–17736”.         

I recommend the work for publication.

Best regard

Reviewer 2 Report

the revised manuscript deserves publications